# G-Protein Coupled Receptors (GPCRs) in Insects—A Potential Target for New Insecticide Development

**DOI:** 10.3390/molecules26102993

**Published:** 2021-05-18

**Authors:** Nannan Liu, Ting Li, Yifan Wang, Shikai Liu

**Affiliations:** 1Department of Entomology and Plant Pathology, Auburn University, Auburn, AL 36849, USA; tzl0001@auburn.edu (T.L.); yzw0093@auburn.edu (Y.W.); 2College of Fisheries, Ocean University of China, Qingdao 266100, China; liushk@ouc.edu.cn

**Keywords:** G-protein coupled receptor regulation pathway, GPCR physiological functions, tissue specific expression, genome sequences analysis, phylogenic tree, insect physiology, insecticide resistance

## Abstract

G-protein coupled receptors (GPCRs) play important roles in cell biology and insects’ physiological processes, toxicological response and the development of insecticide resistance. New information on genome sequences, proteomic and transcriptome analysis and expression patterns of GPCRs in organs such as the central nervous system in different organisms has shown the importance of these signaling regulatory GPCRs and their impact on vital cell functions. Our growing understanding of the role played by GPCRs at the cellular, genome, transcriptome and tissue levels is now being utilized to develop new targets that will sidestep many of the problems currently hindering human disease control and insect pest management. This article reviews recent work on the expression and function of GPCRs in insects, focusing on the molecular complexes governing the insect physiology and development of insecticide resistance and examining the genome information for GPCRs in two medically important insects, mosquitoes and house flies, and their orthologs in the model insect species *Drosophila melanogaster*. The tissue specific distribution and expression of the insect GPCRs is discussed, along with fresh insights into practical aspects of insect physiology and toxicology that could be fundamental for efforts to develop new, more effective, strategies for pest control and resistance management.

## 1. Introduction

G-protein-coupled receptors (GPCRs), which are proteins that share a seven α-helical transmembrane structure, govern a number of biological and physiological processes in both the vertebrate and invertebrate kingdoms. The main function of GPCRs is to transduce extracellular and environmental signals and regulate intracellular second messengers through coupling to heterotrimeric G-proteins and their downstream effectors [1]. GPCRs are known to be involved in recognizing extracellular messengers, transducing signals to the cytosol, and mediating the cellular responses necessary for the normal physiological functions of organisms [2,3,4,5,6,7]. GPCR binding to a wide variety of molecules (ligands), including hormones, neurotransmitters, ions, photons, odorants, neuropeptides and other stimuli through coupling with G proteins and arrestins [1], results in interactions with G proteins that, in turn, activate the downstream effectors of GPCR pathways, including the adenylate cyclase/cyclic AMP (AC/cAMP), phosphatidylinositol/diacylglycerol/protein kinase C (PI/DAG/PKC) and voltage gated calcium channel (Ca^2+^ channels) pathways, ultimately playing an indispensable role in the organism’s growth, development, reproduction and other physiological activities [6,7,8,9,10]. These critical functions mean that a better understanding of the role played by GPCRs in cell physiology and biochemistry is crucial for efforts to develop new molecular-level human disease therapies [10,11,12,13,14,15,16].

In just a few short years, GPCR research in insects has progressed from the initial GPCR gene identification to comprehensive bioinformatics analyses, from single GPCR gene analysis to whole genome sequences of GPCRs and explorations of their pathways, and from traditional transcriptional analysis of the gene expression to gene functional characterization of the GPCR genes in insect physiology and cellular biology. The incredible progress being made in related fields includes a wide range of complementary technologies, including bioinformatics and quantitative expression analyses, with functional studies using RNA interferon revealing potential biological functions that significantly impact insect physiology [17,18], including reproduction [19,20,21,22,23,24], regulating growth and development [21,25,26,27,28,29,30,31,32,33,34,35], the stress response [36,37,38,39,40,41,42,43,44,45,46,47], feeding [21,35,48,49,50,51,52,53,54,55,56], general behaviors [20,57,58,59,60,61,62] and other physiological processes [39,63,64,65,66,67,68,69,70]. In addition, the overexpression of GPCRs in insecticide resistance has been reported in both mosquitoes [42,43,44], and house flies [71,72]. The functions of GPCR regulatory pathways and GPCR downstream intracellular cascades have been explored in the development of insecticide resistance in *Culex quinquefasciatus* [42,43,44], providing a good understanding of the complex molecular processes that govern the development of insecticide resistance through the GPCR signally transduction pathways. The outcomes of these physiological and toxicological studies not only provide us with a clear global picture that is enabling us to develop a clearer understanding of the highly complex mechanisms, genes and pathways involved in these insect physiological and toxicological processes, but are also providing a strong foundation that will allow us to develop tightly GPCR targeted new insecticides and/or environmentally sound insecticides for better vector control that circumvent the problems associated with resistance, representing a highly practical application of scientific research in this area. The findings reported in the most recent studies of GPCR functions in insect are opening up promising new avenues that will undoubtedly revolutionize future research on insect pest management.

This review article examines our current knowledge of the genomic information of the GPCRs in several important insect species and explores the general relationships between and functions of GPCRs in insect physiology. This genome sequencing and annotating information is essential for efforts to build a strong foundation that will enable us to pinpoint the molecular mechanisms and functions of GPCRs in insect physiology and molecular biology in general. We also discuss the distribution and location of GPCRs in various insect tissues where GPCRs are thought to play critical roles in the regulation of insect physiological processes. Since GPCRs are being considered as potential new targets for novel insecticides, understanding the GPCR expression locations and functions could provide valuable insights for those working on developing new insecticides for more effective insect pest control, especially in resistant populations, by identifying possible targets for new chemical and/or insecticide approaches.

## 2. Whole Genome Sequencing and Transcriptome Analysis—Sequence Comparison and GPCR Characterization in Insects

Recent developments in high-throughput sequencing (HTS) technologies have created powerful analytic tools that enable researchers to study the complex gene interactions in individual organisms and the diverse relationships between organisms. Whole-genome analysis and transcriptome gene expression have revolutionized efforts to identify and annotate insect GPCRs in many different insect species, with comparative genomics of insect populations being used to identify novel targets for insect pest control. Given the growing problem of the development of resistance to the insecticides that are conventionally used to control insects, especially those that are medically, agriculturally and economically important [73], there is an urgent need to shift from traditional chemical pesticides towards more targeted gene-centric approaches. Indeed, genome resequencing analysis followed by functional characterization has opened up many opportunities for discovering new targets, such as GPCRs, for new insecticides that not only control insect pests effectively but also overcome the problem of insecticide resistance. This section reviews GPCRs and their possible functions for several insect species at the whole genome level.

### 2.1. Classification Systems Used in Characterizing GPCRs

In general, two classification systems are used to characterize GPCRs in organisms, namely the A-F system [74,75] and GRAFS [76]. The A-F system classifies GPCRs primarily in terms of their sequences and functional similarities using six classes, labelled A, B, C, D, E and F. Class A, known as the “rhodopsin-like family”, is the largest group of GPCRs; Class B is the “secretin receptor family”; Class C contains the metabotropic glutamate family; Class D refers to the fungal mating pheromone receptors; Class E includes all the cAMP receptors; Class F are the frizzled/smoothened receptors [77]. The GRAFS system is based on the phylogenetic tree of the human GPCR sequences, with GPCRs being classified into five families: glutamate (G), rhodopsin (R), adhesion (A), frizzled/taste 2 (F) and secretin (S) [76]. In insects, the classification of GPCRs is based on the A-F system. Work on insect GPDRs was revolutionized by the publication of first insect genome sequence of *Drosophila melanogaster*, an important model organism [78]. To date, more than 100 insect genomes have been sequenced and published, including *Anopheles gambiae* [79], *Aedes aegypti* [80], *Cx. quinquefasciatus* [81], *Musca domestica* [82] and some other species [83]. Sequencing and annotating these insect genomes provide a strong foundation for future research and new insights that enrich our understanding of the potential functions of GPCRs in insect physiology and molecular biology.

As an important model species in the insect kingdom, *D. melanogaster* has 200 GPCR genes, over 70 of which belong to Class A (the rhodopsin-like family), 20 to Class B (the secretin-like family), 5 to Class C (the metabotropic glutamate-like family) and 5 to Class F (the frizzled/smoothened family); the remainder have not yet been clearly classified [84,85]. A homology-based bioinformatics analysis conducted on the genome of the mosquito *An. gambiae* revealed 276 GPCR genes, of which over 80 belong to Class A (the rhodopsin-like family), 20 to Class B (the secretin-like family), 8 to Class C (the metabotropic glutamate-like family), 8 belong to Class F (frizzled/smoothened) and the remainder were other GPCRs [86]. Several GPCRs were explored via reannotation in the mosquito *Ae. aegypti*, coding over 135 GPCR genes, with 89 belonging to Class A (the rhodopsin-like family), 24 to Class B (the secretin-like family), 8 to Class C (the metabotropic glutamate-like family), 11 belonging to Class F (frizzled/smoothened family) and the rest not been clearly classified [80]. Around 90 GPCRs from the silkworm *Bombyx mori* have also been annotated by bioinformatics analysis, with 69 being classified as Class A (the rhodopsin-like family), 8 as Class B (the secretin-like family), 9 as Class C (the metabotropic glutamate-like family), 3 as Class F (the frizzled/smoothened family) and others [87]. For *Apis mellifera*, around 50 genes have been coded for GPCR, of which 31 belong to Class A (the rhodopsin-like family) and 4 to Class B (the secretin-like family) [17]. For *Cx. quinquefasciatus*, around 115 GPCR genes have been identified, with 52 belonging to Class A (the rhodopsin-like family), 4 to Class B (the secretin-like family) and the rest not been classified [42]. For *Musca domestica*, around 94 genes have been identified as GPCR genes, 55 of which are classified as Class A (the rhodopsin-like family), 27 are classified as Class B (the secretin-like family), 4 are classified as Class C (the metabotropic glutamate-like family) and the rest are still unclear [72] (Table 1, Figure 1).

In addition to the insects above that have been extensively studied, GPCRs from several other insect species have also been identified and annotated. The genome of the blowfly, *Lucilia cuprina*, has also been characterized by typical genomic sequencing, RNA-seq and the assembly method. The subsequent genome sequence and gene/protein identification and annotation revealed 197 GPCRs, including 73 to be in Class A (the rhodopsin-like family), 18 in Class B (the secretin-like family) and 9 in Class C (the metabotropic glutamate receptor family), along with some unclassified proteins [88] (Figure 1, Table 1). In recent work, a transcriptome study on GPCRs in the stick insect, *Carausius morosus*, conducted via RNA-seq and de novo RNA assembly revealed 430 putative GPCR genes [89]; a transcriptome analysis predicted over 300 transcripts for coding GPCRs in fire ants [90] and 65 genes for putative neuropeptide GPCRs were identified in the spider mite [91].

Generally, Class A GPCRs are the predominant class of GPCRs in insects, comprising their neuropeptide and protein hormone receptors, muscarinic acetylcholine receptors, dopamine receptors, 5-hydroxytryptamine receptors, tyramine receptors, opsin receptors, vasopressin receptors and orphan receptors, among others [92,93]. These perform a variety of have different functions [94,95,96], including metamorphosis (e.g., dopamine receptors) [29,34,35,97,98], feeding modulation (e.g., allatostatin receptors) [99,100], behavioral changes (e.g., sex peptide receptors and SIF amide neuropeptide receptors) [101,102,103,104], metabolism regulation (e.g., CCHamide-2 receptors and adipokinetic hormone receptor) [51,53,62,105,106] and visual photoreceptors (e.g., rhodopsin proteins) [107], among many others [92,95,108,109,110]. The Class B GPCRs, which include diuretic hormone receptors, methuselah-type receptors and others [111,112], mainly focus on the insect’s water balance (e.g., corticotropin-releasing factor receptors) [113], body temperature modulation (e.g., diuretic hormone receptors) [114], stress response (e.g., methuselah GPCRs) [115,116] and other functions [117]. Their Class C receptors are mainly composed of metabotropic GABA receptors, which serve as neurotransmitters receptors and are involved in signal transportation [118]. The frizzled receptors and smoothened-type receptors, which are mainly Class F GPCRs, focus on Wnt signaling [119,120,121] and the hedgehog signaling pathway, respectively [122,123] (Figure 1). Both the Wnt signaling pathway and the hedgehog pathway are important for insect development [124,125].

### 2.2. Receptors of GPCR Involved in Insect Physiology or Insecticide Resistance That Are Potential Targets for Insecticide Development

Several GPCRs have been reported to be involved in different functions in insect physiology as described below (Table 2) and could thus be potential targets for new insecticides. The majority are Class A GPCRs, including the dopamine receptor, which mainly regulates sexual activity [60,126], morphogenesis [29,34,35,97], mushroom and locomotor activity [127] and ethanol-induced sedation [68]. Neuropeptide and hormone receptors such as the adipokinetic hormone receptor (AKHR) can receive signals from the adipokinetic hormone and regulate lipid mobilization [51,53,62,106], while the allatostatin receptor (AstR) regulates the juvenile hormone synthesis [26,128], the diapause hormone receptor (DHR) is involved in insect development [39,41,45] and the neuropeptide receptors focus on the regulation of insect feeding behavior [50,129] and ecdysone synthesis [33]. Orphan receptor, like BNGR in *B. mori* regulates the insect’s food intake and growth [130], and DLGR2 in *D. melanogaster* regulates the insect’s bursicon bioactivity [25]. Tyramine receptors are responsible for the development of antiparasitic compounds [131], while calcitonin receptors regulate hindgut contraction and primary urine secretion [66] and the 5-HT (5-hydroxytryptamine) receptors and muscarinie acetylcholine receptors are important for the insect’s neural processes [132,133,134]. Rhodopsin receptors perform vital functions in both the insect’s reproduction system [22] and its vision [107]. Class B GPCRs such as methuselah receptors are also involved in insect longevity [135], and oxidative stress resistance [136] and diuretic hormone receptors regulate the body temperature and homeostasis [114]; GPCRs in the Class C family, such as metabotropic GABA receptors, are important for the central nervous system (CNS) [118].

In addition to the vital functions they perform in the insect’s physiology, GPCRs are also involved in insecticide resistance. The methuselah-like receptor in *L. dispar* [135], the calcitonin receptor and opsin receptor in *Cx. quinquefasciatus* [42] and the arrestin gene in *Culex pipiens* [137] have all been shown to be involved in the upregulation of detoxified enzymes such as cytochrome P450. The leucokinin receptor, opsin receptor, methuselah-like receptors and dopamine receptors have also exhibited higher levels of expression in resistant housefly strains compared to susceptible strains [72]. Octopamine and tyramine receptors are also known to be involved in Amitraz resistance in ticks based on the results of an SNP analysis [138].

Chemicals or techniques that target these receptors and destroy their function could serve as very effective insecticides. For example, chemicals that act on dopamine receptors like amitriptyline and doxepin have been shown to cause the death of both *Ae. aegypti* [139,140] and *Cx. quinquefasciatus* [141], while an RNAi technique that targets the dopamine 1 receptor also led to the death of both larvae and adult in *Ae. aegypti* [142]. Many more chemicals could be investigated to target various GPCR receptors and improve insect management (Table 3).

## 3. Tissue Specific Expression Analysis of GPCRs in Insects

Tissue specific analyses of gene expression usually provide new insights into potential physiological mechanisms and correlates the functions of the genes with the functions of specific tissues. GPCRs are distributed in various insect tissues, where GPCRs and their ligands play critical roles in the regulation of insect physiological processes. Since the focus here is on considering GPCRs as new targets of novel insecticides, understanding the GPCR expression locations and functions could provide valuable insights and contributions for new insecticide development. Discovering and identifying the GPCR gene profiles in different tissues will help broaden our understanding of the functions of various types of tissue and gene functions, and the biological mechanisms involved [143]. The aberrant expression of tissue specific expressed GPCRs may also be implicated in various abnormal functions of the insects, and hence important for the control of insect pests. This section reviews recent discoveries related to the GPCR genes present in insect nervous, digestive and reproductive systems, and in others, such as fat bodies, muscles and salivary glands (Figure 2).

### 3.1. Brain Tissue and Central Nervous System

Several GPCRs have been identified as being predominately expressed in the brain and CNS, corresponding to neuropeptide expression in diverse insect species. A neuropeptide bona fide natalisin receptor found to be highly expressed in the CNS of *D. melanogaster*, *T. castaneum* and *B. mori* regulates sexual activity and fecundity in insects [20]. A mutant *methuselah* gene expressed in the brain of *Drosophila*, a model insect often used to investigate gene functions, plays a critical function in oxidative stress resistance [136]; a dopamine/ecdysteroid receptor that is a head-specific expression gene is also overexpressed in *Drosophila* adults and embryos [97]. Another study also found the dopamine/ecdysteroid receptor (*DopEcR*) to be highly expressed in *Drosophila* nervous system and brain tissues, and its mutant has been implicated in the insect’s resistance to alcohol [68]. A *Drosophila* sex/myp-inhibiting peptide receptor expressed in the CNS is highly conserved in insects [19], while an octopamine receptor *DmOctα2R* transcript present at much higher levels in *Drosophila* males than in females is mainly expressed in the head of adults [144] and a *Drosophila* neuropeptide capa GPCR expressed in larvae central nerve system is responsible for sensitivity to desiccation stress [37]. Of two lGRs (*LGR3/LGR4*) characterized in diverse *Drosophila* tissues, *Lgr3* transcripts were predominately expressed in the CNS, while *Lgr4* transcripts were highly expressed in the CNS of the wandering larval stage. In adults, the *Lgr4* transcripts showed significantly high expression in the male thoracic-abdominal ganglion and brain tissues [145].

The silkworm, *B. mori*, which is classed as a beneficial insect, is another model species frequently used in gene functional studies. A neuropeptide GPCR A4 receptor gene (*BomNPFR*) amplified from its brain tissue was predicted to be involved in food intake and growth processes [130]. Three GPCR receptors responding to neuron ion transport peptides (ITPs) were identified in silkworm larvae using tissue specific expression, revealing that bngr-A2 was mainly expressed in the brain, with no expression in the CNS, and bngr-A34 was not present in nervous tissues [21]. A neuropeptide allatotropin receptor was significantly overexpressed in the corpora cardiaca and brain of *B. mori*, indicating the potential involvement of juvenile hormone (JH) biosynthesis processes [146]. In honeybees, an important beneficial insect that plays a critical role in pollination, an octopamine receptor was identified in the honeybee brain [63]. A tyramine receptor was later identified that was highly expressed in nurse bees and forager brain neuropils [147].

The red import fire ant, *Solenopsis invicta*, is an invasive urban insect species whose gene expression and functions have been widely studied over the last couple of decades. Ten GPCR gene expressions have exhibited significant differences in the brain tissue of workers, alate virgin queens, and mated queens [90]. One neuropeptide F-Like receptor was isolated from *S. invicta* and its differential expression levels in the brain suggest its potential function to be feeding regulation for mated queens [48]. In the medically important kissing bug, *Rhodnius prolixus*, an important Chagas disease vector, GPCRs have been found to be involved in multiple physiological pathways. Three variants of a pyrokinin-1 receptor gene characterized in different tissues of fifth instar larvae of *R. prolixus* revealed two variants that were mainly expressed in the CNS [148]. A corazonin receptor was found to be overexpressed in the brain and involved in *R. prolixus* heartbeat control [149] and a serotonin type 2b receptor transcript has also been shown to be upregulated in the CNS of *R. prolixus* [150]. To predict the potential function of a kinin receptor in *R. prolixus* blood post-feeding, a kinin receptor was characterized in various tissues and found to be overexpressed in the CNS of late instar larvae [56]. In mosquitoes, which are responsible for transmitting a number of human and animal diseases, GPCRs have been identified as being involved in mosquito blood feeding and insecticide resistance. The overexpression of an allatotropin GPCR receptor (*AeATr*) gene was characterized in the nervous system and corpora alata-corpara cardiac complex of *Ae. aegypti*. Blood feeding depressed the transcript level of *AeATr*, and was associated with JH biosynthesis in mosquitoes [32]. A rhodopsin-like GPCR overexpressed in the head of adult mosquitoes was found to play a critical role in the development of permethrin insecticide resistance in the mosquito, *Cx. Quinquefasciatus* [43]. Tissue specific studies conducted on this GPCR gene revealed its significantly high expression in the insect’s brain tissue along with a G-protein alpha subunit, two adenylyl cyclase and one protein kinase A gene, playing important roles in the GPCR-leading intracellular pathway in the insect’s neuron system and regulating its insecticide resistance [44]. A corazonin neuropeptide receptor (*MdCrz*) has been found to be overexpressed in the larval CNS of the house fly, another medically important pest, including in its ventral nerve cord, the protocerebral DL neurons of the brain lobes and its *vCrz* neurons. Although it was overexpressed in the heads of both male and female houseflies, there was no expression in other body tissues, indicating that *MdCrz* is a brain-specific expression gene [151].

The red flour beetle, *T. castaneum*, is a stored product pest that causes problem for communities around the world. The spatial expression of a 5-HT7-type serotonin receptor identified in *T. castaneum* showed the highest expression levels to be in the head of both male and female adults, predominantly expressed in the brain but accompanied by high levels of expression in the optic lobes, predicting the functional importance of this receptor in neural processes [134]. A D2-like dopamine receptor gene predominantly expressed in the head and CNS of *T. castnaeum* adults [152] and an inotocin receptor was found to be overexpressed in the head of the insect’s early larval stage [153]. In the desert locust, *S. gregaria*, an agricultural pest species, two novel octopamine receptors were characterized in adults, with *SgOctαR* being overexpressed in the CNS, including the brain, optic lobes, subesophageal ganglion and thoracic ganglions, while *SgOctβR* was overexpressed in the CNS. Interestingly, these receptors were highly expressed in the long-term gregarious locusts but not in the solitarious locusts, suggesting the receptors’ function in locust behavior [154]. GPCRs have also been widely studied in other agricultural pest species. The GPCR *NIA42* was found to be highly expressed in the brain and abdominal integuments of the adult brown planthopper, *N. lugens*, where it was linked to the neuropeptide NI-elevenin that is involved in the regulation of planthopper melanization [70,155]. A novel octopamine receptor gene expression has been characterized in diverse tissues of the fifth-instar larva rice stem borer, *Chilo suppressalis*, showing a high level of expression in the nerve cord [156]. A large screen of GPCRs that was characterized in *C. suppressalis* identified 51 putative GPCR genes. The expression of these genes was examined in tissues of the insect’s CNS, fat body, gut and hemocytes, with most of the receptor genes being highly expressed in the CNS [157]. A serotonin receptor (*Pr5-HT_8_*) was identified as being highly expressed in the nerve cord of the larva small white butterfly, *Pieris rapae*. 5-hydroxytryptamine (5-HT) has been identified as a neurotransmitter that plays critical roles in the regulation of a number of physiological processes, which is consistent with the expression of *Pr5-HT8* in the butterfly’s CNS [133]. An allatotropin receptor has also been found to be overexpressed in the brain, thoracic ganglion and abdomen ganglion of the bollworm, *H. armigera* [158].

### 3.2. Digestion and Reproduction Systems

In mosquito species, GPCRs are known to be expressed in the digestion system and involved in blood-feeding behavior. A leucokinin receptor has diverse functions, responding to multiple kinins in the mosquito, *Ae. aegypti*, and is expressed in the mosquito’s hindgut and Malpighian tubules [159]. Piermarini’s group also identified several GPCRs and metabolic genes upregulated or downregulated in the Malpighian tubules of blood-taken *Aedes albopictus* [160]. An allatotropin GPCR receptor (*AeATr*) gene has also been found to be overexpressed in the ovary of adult mosquitoes [32]. In another blood-feeding insect, *R. prolixus*, a serotonin type 2b receptor transcript was overexpressed in the Malpighian tubules, salivary glands and guts [150]. A kinin receptor has also been found to be highly overexpressed in the gut system of late instar larvae of *R. prolixus* [56]. Two variants of a pyrokinin-1 receptor gene were mainly expressed in the male testes and prothoracic glands of fifth instar larvae of *R. prolixus* [148]. In *Drosophila*, several GPCRs were identified as overexpressed in the digestion and reproduction systems. These *Drosophila Lgr4* transcripts were highly expressed in the gut system of the wandering larval stage, and significantly highly expressed in the male midgut and crop [145]. Additionally, a sex/myp-inhibiting peptide receptor was highly expressed in the male *Drosophila* reproduction organs [19] and an octopamine receptor gene was highly expressed in the Malpighian tubules, with lesser amounts found in the midgut and hemocytes of the fifth-instar larva rice stem borer, *C. suppressalis* [156]. Several GPCRs in *C. suppressalis* were overexpressed in the gut system [157]. A neuropeptide F-Like receptor detected in the gut and reproduction systems of *S. invicta* suggests its potential function was involved in the feeding regulation of mated queens [48]. One ITP gene identified in *B. mori*, bngr-A2, was found to be mainly expressed in the reproduction system [21]. A serotonin receptor (*Pr5-HT_8_*) is known to be highly expressed in the Malpighian tubules, fat body and midgut of larvae of the small white butterfly, *P. rapae* [133].

### 3.3. Other Insect Organs

In addition to the GPCRs identified in the CNS, digestive and reproductive systems in insect species, many GPCRs have also been characterized in other organs. In *Drosophila* adults, Lgr3 transcripts were found to be highly expressed in the female fat body and uterus, and the male salivary glands [145]. An adipokinetic hormone receptor was predominately expressed in the fat body of the oriental fruit fly, *B. dorsalis*, and was also involved in its triacylglycerol mobilization and sexual behavior [62]. Forty six putative GPCR transcripts have been isolated from the foreleg tissues of the cattle tick, *Phipicephalus australis*, providing valuable information for GPCR studies of signal transduction, host preference and mating behavior in insects [161]. An allatotropin receptor has been identified that is overexpressed in the male bumblebee accessory glands, predicting its potential involvement in JH biosynthesis [162] and in *B. mori* a sex peptide receptor has been found to be highly expressed in the prothoracic gland, predicting the critical roles of receptor in regulating ecdysteroidogenesis [28]. A novel octopamine receptor, *SgOctβR*, was also overexpressed in the flight muscles of *S. gregaria*, suggesting its involvement in locust gregarious behavior [154], while a GPCR receptor, *NlA42*, was significantly highly expressed in the integument and salivary gland of the brown planthopper indicating its function in the insect’s melanization [155].Figure 2Tissue specific expression of GPCR genes in insect species. Insect GPCRs are mainly expressed in the brain and central nervous system, highlighted in light blue; those in the digestive system are highlighted in gray; those in the reproductive system are highlighted win light purple; those in other organs are highlighted in orange. Twenty-six GPCRs that are highly expressed in the brain and central nervous system in 13 insect species are involved in sexual activity and fecundity, oxidative stress resistance, desiccation stress, food intake and growth, juvenile hormone biosynthesis, feeding regulation, heartbeat control, permethrin insecticide resistance, behavior and cuticle melanization.
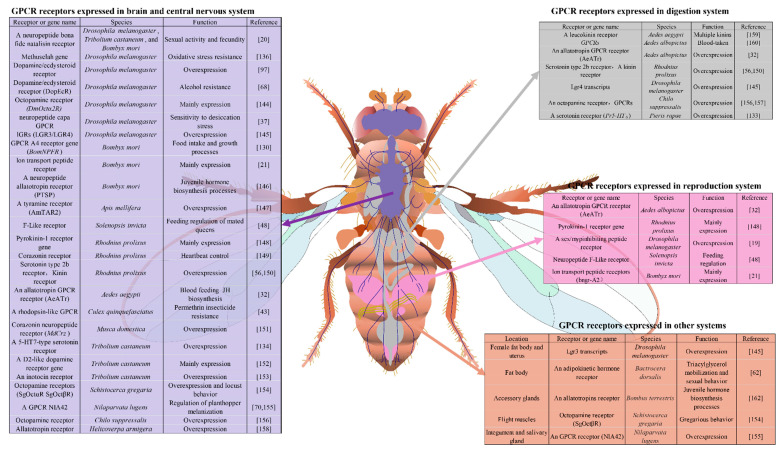


However, while the precise functions of the GPCRs in specific tissues are still in need of further characterization, the impact of the GPCRs on insect behaviors can now be predicted to some extent based on the functions of those specific tissues in insects. The importance of GPCRs in insects can be partially appreciated by considering their distribution, location and function within the cells of specific tissues. In humans, if any of the fundamental processes governed by GPCRs in specific tissues are dysfunctional, this will lead to acute or chronic diseases [163,164,165,166]. The physical location and expression of GPCRs may provide a direct mechanism for the transduction of extracellular messages into tissue responses and functions. Consequently, the identification of the GPCRs in brains and CNS of insects could shed new light on the important roles they play in neuronal firing, the regulation of ion transportation across cell membranes and the modulation of neuron membrane potentials in other species.

## 4. Conclusions

The biological and medical importance of GPCRs has been well characterized and is now widely recognized, making GPCR an important focus for drug discovery. Yet, while none or few insecticides have been developed that specifically target GPCRs in insects, recent progress on identifying the GPCR genome sequences in insects, the specific expression of the GPCRs in different tissues and the crucial role of GPCRs for insect physiologic processes and toxicology promise to provide fresh insights into GPCRs’ biochemical functions in insects and assist in the development of new insect-specific insecticides. The value of the findings summarized in this review of the current status of research into GPCR in insects, including information on their sequences, functions in physiology and toxicology and the tissues and organs that they are involved in, are fundamental for understanding how these GPCR systems function to modulate a broad spectrum of cellular activities. A recent review on insect GPCRs uncovered GPCR signaling pathways, functions in insect physiology and toxicology and the latest exciting technological advances and new techniques for gene expression and functional of the GPCRs in insects [167]. Together, these reviews on insect GPCRs will help researchers in the field develop new insect-specific insecticides that will help control insect pests in the future.

## Figures and Tables

**Figure 1 molecules-26-02993-f001:**
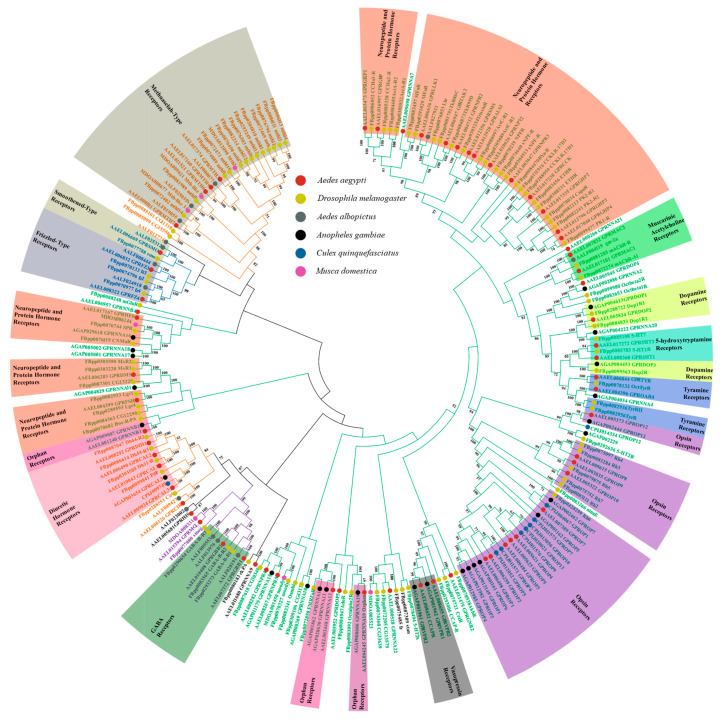
Sequence comparison of GPCRs in insects. The phylogenetic tree includes 64 *Ae. aegypti* GPCRs, 98 *D. melanogaster* GPCRs, 14 *Aedes albopictus* GPCRs, 26 *An. Gambiae* GPCRs, 9 *Cx. Quinquefasciatus* GPCRs and 7 *M. domestica* GPCRs. Different colored circles represent different species. Except for the genes listed for *D. melanogaster*, which are taken from Flybase (https://flybase.org, accessed on 7 May 2021), all the genes are from Vectorbase (https://vectorbase.org, accessed on 7 May 2021). Sequence alignment was conducted in MEGAX and the phylogenetic tree was developed using a neighbor-joining method by MEGAX with 2000 bootstrap replicates. The evolutionary distances were computed using the p-distance method. Different classes of GPCRs are represented by colored branches—green for the Class A (the rhodopsin-like family), yellow for the Class B (the secretin receptor family), purple for the Class C (the metabotropic glutamate family) and blue for the Class F (the frizzled and smoothened receptor family). The major types of GPCRs in each of classes are labeled.

**Table 1 molecules-26-02993-t001:** The information of GPCR genes in insect species.

Insect Species	Total Number of Genes	Class A Gene Number	Class B Gene Number	Class C Gene Number	Class F Gene Number	Source of Genome Information	Reference
*D. melanogaster*	~200	>70	~20	~5	~5	https://flybase.org/ (accessed on 7 May 2021)	[84,85]
*An. gambiae*	~276	81	21	8	~8	https://www.ncbi.nlm.nih.gov/genome/46?genome_assembly_id=22679 (accessed on 7 May 2021)	[86]
*Ae. aegypti*	135	89	24	8	11	https://www.ncbi.nlm.nih.gov/genome/44?genome_assembly_id=322291 (accessed on 7 May 2021)	[80]
*B. mori*	~90	~70	~7	~8	~4	https://www.ncbi.nlm.nih.gov/genome/76?genome_assembly_id=1491718 (accessed on 7 May 2021)	[87]
*A. mellifera*	~50	~31	~4	Not clear	Not clear	https://www.ncbi.nlm.nih.gov/genome/48?genome_assembly_id=403979 (accessed on 7 May 2021)	[17]
*L. cuprina*	197	73	18	9	Not clear	https://www.ncbi.nlm.nih.gov/genome/12732?genome_assembly_id=358015 (accessed on 7 May 2021)	[88]
*Cx. quinquefasciatus*	115	52	4	Not clear	Not clear	https://www.ncbi.nlm.nih.gov/genome/393?genome_assembly_id=1502880 (accessed on 7 May 2021)	[42]
*M. domestica*	94	55	27	4	Not clear	https://www.ncbi.nlm.nih.gov/genome/14461?genome_assembly_id=44793 (accessed on 7 May 2021)	[72]

**Table 2 molecules-26-02993-t002:** The functions of GPCR receptors in insect physiology.

Receptor Group	Receptor Name	Classes	Species	Function	Reference
5-HT receptors	Trica5-HT7 R	Class A	*Tribolium castaneum*	Insect’s neural processes	[134]
Adipokinetic hormone receptor	Akh receptor	Class A	*Bactrocera dorsalis*	Lipid mobilization	[62]
Adipokinetic hormone receptor	Akh receptor	Class A	*D. melanogaster*	Lipid mobilization	[51]
Adipokinetic hormone receptor	Akh receptor	Class A	*Nilaparvata lugens*	Lipid mobilization	[53]
Allatostatin receptor	AstAR1	Class A	*D. melanogaster*	Metamorphosis	[98]
Allatostatin receptor	DAR-1/DAR-2	Class A	*D. melanogaster*	Feeding modulation	[99]
Allatostatin receptor	Dippu-AstR	Class A	*Diploptera punctata*	Juvenile hormone synthesis	[128]
Arginine vasopressin-like receptor	AVPL receptor	Class A	*T. castaneum*	Diuretic signaling pathway	[92]
Calcitonin receptors	GPCRCAL1	Class A	*Ae. aegypti*	primary urine secretion	[66]
CCHa2 receptor	CCHa2-R	Class A	*D. melanogaster*	Insulin production	[105]
Diapause hormone receptor	DH-R	Class A	*Ae. aegypti*	Development	[39]
Diapause hormone receptor	Bommo-DHR	Class A	*B. mori*	Development	[45]
Diapause hormone receptor	HzDHr	Class A	*Helicoverpa zea*	Development	[41]
Dopamine receptor	Dop1R2, DmDopEcR	Class A	*D. melanogaster*	Morphogenesis	[34,97]
Dopamine receptor	DopEcR	Class A	*D. melanogaster*	Mushroom and locomotor activity	[127]
Dopamine receptor	DopEcR	Class A	*D. melanogaster*	Ethanol-induced sedation	[68]
Dopamine receptor	AipsDopEcR	Class A	*Agrotis ipsilon*	Sexual activity regulation	[60,126]
Dopamine receptor	DopEcR	Class A	*Helicoverpa armigera*	Morphogenesis	[35]
Dopamine receptor	D2R	Class A	*T. castaneum*	Morphogenesis	[29]
Leucokinin receptor	LKr	Class A	*D. melanogaster*	Feeding modulation	[100,103]
Myosuppressin receptors	CG8985/CG13803	Class A	*D. melanogaster*	visceral muscles inhibition	[108]
Neuropeptide receptors	GPCR-B2	Class A	*B. mori*	Ecdysone synthesis	[33]
Neuropeptide receptors	Schgr-sNPFR	Class A	*Schistocerca gregaria*	Feeding behavior	[50]
Neuropeptide Drosulfakinin receptor	CCKLR-17D1	Class A	*D. melanogaster*	Fighting behavior	[104]
Orphan receptor	DLGR2	Class A	*D. melanogaster*	Bursicon bioactivity	[25]
Orphan receptor	BNGR-A4 receptor	Class A	*B. mori*	Food intake and growth	[130]
Rhodopsin receptors	Rh2	Class A	*T. castaneum*	Reproduction	[22]
Sex peptide receptor	SPR	Class A	*D. melanogaster*	Reproductive behavior	[101]
SIFamide receptor	SIFaR	Class A	*D. melanogaster*	Reproductive behavior	[102]
Tyramine receptor	TAR1	Class A	*Rhipicephalus (Boophilus) microplus*	Development of antiparasitic	[131]
Corticotropin releasing factor receptor	CG12370	Class B	*D. melanogaster*	Water balance	[113]
Diuretic hormone receptors	DH31R	Class B	*D. melanogaster*	temperature regulation and homeostasis	[114]
Methuselah receptor	mth	Class B	*D. melanogaster*	Oxidative stress resistance	[136]
Methuselah receptor	Ldmthl1	Class B	*Lymantria dispar*	Insect longevity	[135]
Metabotropic GABA receptors	D-GABABR1, R2 and R3	Class C	*D. melanogaster*	Central nervous system	[118]

**Table 3 molecules-26-02993-t003:** The GPCR genes that been reported in insecticide resistance.

Receptor Name	Gene	Class	Species	Insecticide	References
Calcitonin receptor	CPIJ014419	Class A	*Cx. quinquefasciatus*	Permethrin	[42]
Pteropsin	CPIJ014334	Class A	*Cx. quinquefasciatus*	Permethrin	[42]
Conserved hypothetical protein	CPIJ019111	Not clear yet	*Cx. quinquefasciatus*	Permethrin	[42]
Leucokinin receptor	LOC101891982	Class A	*M. domestica*	Imidacloprid	[72]
Opsin receptor	LOC101900880, LOC101900148	Class A	*M. domestica*	Imidacloprid	[72]
Methuselah-like receptor	LOC101889292, LOC101899380, LOC105262457, LOC101894839	Class B	*M. domestica*	Imidacloprid	[72]
Dopamine receptor	LOC101896361	Class A	*M. domestica*	Imidacloprid	[72]
Crustacean cardioactive peptide receptor	LOC101898141	Class A	*M. domestica*	Imidacloprid	[72]
Methuselah-like GPCR	Ldmthl1	Class B	*L. dispar*	Deltamethrin	[135]
Arrestin	HQ833831		*Cx. pipiens*	Deltamethrin	[137]

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
