# Peer review of "G-Protein Coupled Receptors (GPCRs) in Insects—A Potential Target for New Insecticide Development"

_molecules, 2021, doi:10.3390/molecules26102993_

Round 1
Reviewer 1 Report
This review article provides a comprehensive summary of the knowledge of GPCRs in insects, from their classification into GPCRs to the full information on the GPCR genes found in each insect. This paper is like an encyclopedia of insect GPCRs. However, if I were to complain, I would say that the review is inadequate in that it describes the fact that the genes were discovered in various insects in a straightforward list, and there is almost no mention of the molecular mechanisms such as the functions of GPCRs and their effects on the signal transduction system. It would be easier to get an overview if the list of genes were summarized in a table for each species. I think that the story needs to be developed a little more so that the reader can understand it.
Author Response
Review #1:
Comments: This review article provides a comprehensive summary of the knowledge of GPCRs in insects, from their classification into GPCRs to the full information on the GPCR genes found in each insect. This paper is like an encyclopedia of insect GPCRs. However, if I were to complain, I would say that the review is inadequate in that it describes the fact that the genes were discovered in various insects in a straightforward list, and there is almost no mention of the molecular mechanisms such as the functions of GPCRs and their effects on the signal transduction system. It would be easier to get an overview if the list of genes were summarized in a table for each species. I think that the story needs to be developed a little more so that the reader can understand it.
Answer: Thanks for the reviewer’s valuable comments (the reviser #2 has the similar suggestion for adding a Table for summary). We have developed 3 new Tables (Tables 1-3) in this revision to highlight the GPCR genome (classes) information in each insect species (Table 1); the function of the GPCRs in insect physiological functions (Table 2), and GPCRs research in insecticides and/or resistance (Table 3). We believe that these tables make clear summaries for the information relating the GPCRs in insects.
In addition, since a very recent review on insect GPCRs has uncovered GPCR signaling pathways, functions in insect physiology and toxicology, and latest exciting technological advances and new techniques for gene expression and functional of the GPCRs in insects, we will not duplicate the information. But we referred this information in the conclusion section and adding the citation [167].
Reviewer 2 Report
In this little lovely review, Dr. Liu et al., provided the comprehensive genomic information of the GPCRs in several important insect species and explored the general relationships and physiological functions of GPCRs in insect. The tissue specific distribution and expression of them were also discussed. Overall, this review refreshes our insights into the roles played by GPCRs in the insect physiology and toxicology and those works/efforts on developing new insecticides for more effective pest control and resistance management strategies.
I only have several comments/suggestions may can help improve the manuscript, please see below.
- The section numbers are confused and line numbers are missed.
- In “Whole genome sequencing…” section, I would like to see the genome access number or dataset link of the most important insect species discussed in this manuscript.
- The resolution of figure 1 was pretty low and we can’t read it very well. I recommend using vector image instead of bitmap. And font size in figure 1 legend should be consistent.
- P5 paragraph 1-last sentence, “…GPCRs were identified in the spider mite (figure 1).” Was this really showed in fig 1?
- In P5 paragraph 2, the representative receptors in each Class were listed and the Classes were also labeled in figure 1. Can you also label the major Classes (such as rhodopsin-like family, secretin-like family…) in Fig 1?
- In P6 paragraph 1-last sentence, “… are important for the CNS (figure 1).” I don’t think the figure 1 showed this information. Can you make another valuable figure/table with these info? Such as the function of specific GPCRs in different insect species? Including receptor name, which class it is, the physiological function in certain insect, which kind of insecticide was related to this receptor, refs, ect. (Perhaps the table can be combined to figure 2 with the expression column). In don’t think any paragraphs in the section “receptors of GPCRs involved in insect physiology or…” can cite figure 1.
- In the end of section “receptors of GPCRs involved in insect physiology or…”, if it is possible, I would like to see more discussion of recent CRISPR knockout and RNAi knockdown of some certain insect GPCRs, the in vivo studies may provide more information of the physiological function and toxicology of the GPCRs.
Author Response
Review #2:
In this little lovely review, Dr. Liu et al., provided the comprehensive genomic information of the GPCRs in several important insect species and explored the general relationships and physiological functions of GPCRs in insect. The tissue specific distribution and expression of them were also discussed. Overall, this review refreshes our insights into the roles played by GPCRs in the insect physiology and toxicology and those works/efforts on developing new insecticides for more effective pest control and resistance management strategies.
I only have several comments/suggestions may can help improve the manuscript, please see below.
Answer: thanks for the reviewer’s positive comments and careful review, which helped improve the manuscript. Please find our answer to specific comments below.
Comment #1: The section numbers are confused and line numbers are missed.
Answer: The text has been carefully checked and corrected.
Comment #2: In “Whole genome sequencing…” section, I would like to see the genome access number or dataset link of the most important insect species discussed in this manuscript.
Answer: Thanks for the reviewer’s valuable comments (the reviser #1 has the similar suggestion for adding a Table for summary). We have developed 3 new Tables (Tables 1-3) in this revision to highlight the GPCR genome (classes) information in each insect species (Table 1); the function of the GPCRs in insect physiological functions (Table 2), and GPCRs research in insecticides and/or resistance (Table 3). We believe that these tables make clear summaries for the information relating the GPCRs in insects.
Comment #3: The resolution of figure 1 was pretty low and we can’t read it very well. I recommend using vector image instead of bitmap. And font size in figure 1 legend should be consistent.
Answer: Thanks for the comments. Accordingly, we have re-making the figure using vector image. The font size of figure legends is now consistent.
Comment #4: P5 paragraph 1-last sentence, “…GPCRs were identified in the spider mite (figure 1).” Was this really showed in fig 1?
Answer: It is not show in the figure 1, it has been corrected.
Comment #5: In P5 paragraph 2, the representative receptors in each Class were listed and the Classes were also labeled in figure 1. Can you also label the major Classes (such as rhodopsin-like family, secretin-like family…) in Fig 1?
Answer: Agree. We have made it clear in the figure legend of Figure 1. The major types of GPCRs in each of classes are labeled .
Comment #6: In P6 paragraph 1-last sentence, “… are important for the CNS (figure 1).” I don’t think the figure 1 showed this information. Can you make another valuable figure/table with these info? Such as the function of specific GPCRs in different insect species? Including receptor name, which class it is, the physiological function in certain insect, which kind of insecticide was related to this receptor, refs, etc. (Perhaps the table can be combined to figure 2 with the expression column). In don’t think any paragraphs in the section “receptors of GPCRs involved in insect physiology or…” can cite figure 1.
Answer: Agree. Again, as we respond in comment #2, we have developed 3 new Tables (Tables 1-3) in this revision to highlight the GPCR genome (classes) information in each insect species (Table 1); the function of the GPCRs in insect physiological functions (Table 2), and GPCRs research in insecticides and/or resistance (Table 3). We believe that these tables make clear summaries for the information relating the GPCRs in insects.
Comment #7: In the end of section “receptors of GPCRs involved in insect physiology or…”, if it is possible, I would like to see more discussion of recent CRISPR knockout and RNAi knockdown of some certain insect GPCRs, the in vivo studies may provide more information of the physiological function and toxicology of the GPCRs.
Answer: Thanks to the reviewer’s comments. Since a very recent review on insect GPCRs has uncovered GPCR signaling pathways, functions in insect physiology and toxicology, and latest exciting technological advances and new techniques for gene expression and functional of the GPCRs in insects, we will not duplicate the information. But we referred this information in the conclusion section and adding the citation [167].
Round 2
Reviewer 1 Report
I went through the revised manuscript, but the new tables are not in place. Also, in the reply comment, the authors mentioned citation [167] was added as a new reference in the conclusion section, but I can't find the corresponding part. Since the revision points could not find, I cannot judge the revised manuscript properly.
Author Response
Dear Reviewer: I am sorry; I am not sure if you did not get the revision version. Here, I am resubmit the revised version with all corrections, including the three new tables and citation of [167].
Round 3
Reviewer 1 Report
I have reviewed the corrections made by the authors. I can judge that the authors have appropriately responded to the comments.